# Heterologously Expressed Cellobiose Dehydrogenase Acts as Efficient Electron-Donor of Lytic Polysaccharide Monooxygenase for Cellulose Degradation in *Trichoderma reesei*

**DOI:** 10.3390/ijms242417202

**Published:** 2023-12-06

**Authors:** Muhammad Adnan, Xuekun Ma, Yanping Xie, Abdul Waheed, Gang Liu

**Affiliations:** Shenzhen Key Laboratory of Microbial Genetic Engineering, College of Life Sciences and Oceanography, Shenzhen University, Shenzhen 518060, China; alvi.adnan@szu.edu.cn (M.A.); 2060251025@email.szu.edu.cn (X.M.); 2060251042@email.szu.edu.cn (Y.X.);

**Keywords:** CDH, electron donor, LPMO, cellulase, *Phanerochaete chrysosporium*, *Trichoderma reesei*

## Abstract

The conversion of lignocellulosic biomass to second-generation biofuels through enzymes is achieved at a high cost. Filamentous fungi through a combination of oxidative enzymes can easily disintegrate the glycosidic bonds of cellulose. The combination of cellobiose dehydrogenase (CDH) with lytic polysaccharide monooxygenases (LPMOs) enhances cellulose degradation in many folds. CDH increases cellulose deconstruction via coupling the oxidation of cellobiose to the reductive activation of LPMOs by catalyzing the addition of oxygen to C-H bonds of the glycosidic linkages. Fungal LPMOs show different regio-selectivity (C1 or C4) and result in oxidized products through modifications at reducing as well as nonreducing ends of the respective glucan chain. *T. reesei* LPMOs have shown great potential for oxidative cleavage of cellobiose at C1 and C4 glucan bonds, therefore, the incorporation of heterologous CDH further increases its potential for biofuel production for industrial purposes at a reduced cost. We introduced CDH of *Phanerochaete chrysosporium* (*Pc*CDH) in *Trichoderma reesei* (which originally lacked CDH). We purified CDH through affinity chromatography and analyzed its enzymatic activity, electron-donating ability to LPMO, and the synergistic effect of LPMO and CDH on cellulose deconstruction. The optimum temperature of the recombinant *Pc*CDH was found to be 45 °C and the optimum pH of *Pc*CDH was observed as 4.5. *Pc*CDH has high cello-oligosaccharide k_cat_, K_m,_ and k_cat_/K_m_ values. The synergistic effect of LPMO and cellulase significantly improved the degradation efficiency of phosphoric acid swollen cellulose (PASC) when CDH was used as the electron donor. We also found that LPMO undergoes auto-oxidative inactivation, and when *PcCDH* is used an electron donor has the function of a C1-type LPMO electron donor without additional substrate increments. This work provides novel insights into finding stable electron donors for LPMOs and paves the way forward in discovering efficient CDHs for enhanced cellulose degradation.

## 1. Introduction

Biofuel production is vital for worldwide energy demands and provides economic and environmental sustainability [1]. Lignocellulosic biomass serves as an ample source for second-generation biofuel production through enzymatic degradation; however, the cost of cellulose-degrading enzymes is a major barrier to the economical production of biofuels [2]. Fungi are very important players in terrestrial environments for cellulose degradation, and their glycoside hydrolases have been studied more extensively [3]. Great efforts have been made for the characterization and optimization of these cellulases, however, their catalytic activity towards recalcitrant lignocellulose has remained relatively low and proved a major challenge [4]. Different transcriptomic and proteomic analyses have identified various oxidative enzymes of fungi that are involved in the degradation of lignocellulose [5].

*Trichoderma reesei* is an important filamentous fungus known for industrial cellulase production, however, it lacks native cellobiose dehydrogenase (CDH) [6]. Genetic modifications in *T. reesei* QM6a led to improved strains such as QM9414 and Rut-C30 with enhanced cellulase and heterologous protein production [7]. Therefore, we introduced the CDH of *Phanerochaete chrysoporium* (*Pc*CDH) in *T. reesei* QM9414 and observed the synergistic effect of heterologous CDH and C1 LPMO on cellulose degradation. CDH is an oxidoreductase that catalyzes the oxidation of longer cello-oligosaccharides or cellobiose to respective 1-5-δ- lactones [8]. These lactones are hydrolyzed enzymatically by lactonases or spontaneously in solution to aldonic acids [9]. CDH is an extracellular protein that contains two domains, an N-terminal heme-binding cytochrome domain (CYT) and a C-terminal dehydrogenase domain (DH) assisted by flavin [10]. The Flavin domain is ubiquitous across various organisms, being a constituent of the glucose-methanol-choline oxidoreductase superfamily. In contrast, the heme domain and its homologs are exclusive to fungi [11]. Oxidation of cellobiose occurs in the flavin domain with subsequent electron transfer to the heme domain [12]. This reduced heme domain can cause the reduction of a wide range of substances such as organic dyes, metal ions, and quinones as well as molecular oxygen. Quinones can also be reduced at the isolated DH domain (e.g., 1,4-benzoquinone, 2,6-dimethyl-1,4-BQ, etc.). The cytochrome domain is not involved in their reduction. The same is true for DCIP while cytochrome *c* can only be reduced via the cytochrome domain [13].

Lytic polysaccharide monooxygenases (LPMOs) have auxiliary activities (AA) and assist cellulases during cellulose degradation [14]. LPMOs are divided into 8 families depending upon their characterization as described in the CAZy database (AA9-AA11, AA13-AA17) [15,16]. The AA9 family of LPMOs belongs to fungi [17]. These LPMOs in synergism with cellulases can accelerate hydrolysis through oxidative cleavage of crystalline cellulose [14]. Although this synergistic effect is well documented certain factors affect the stability and long-term activity of LPMOs [18]. One such problem is the autoxidative deactivation of LPMOs while using reducing agents like ascorbic acid (Asc), gallic acids, or pyrogallol as electron donors [19]. An exogenous electron donor is required by LPMOs during the reduction of divalent copper ions for activation (copper is the central element of the LPMOs) [19]. Exogenous electron donors are divided into three categories: the first one is the small reducing molecules, the second category includes the oxidoreductases with flavin as a cofactor, and the third category involves the light-sensitive organic and inorganic substances [8,20,21]. CDH belongs to the second category among these electron donors [22]. CDH can transfer the electrons obtained during the oxidation of the substrates from the DH domain to the CYT domain electrons through inter-domain electron transfer (IET), and then transfer the electrons from the CYT domain to LPMO through single electron-transfer (ET) [23,24,25]. CDH possesses the following characteristics for being an effective electron donor of AA9 family LPMOs: (1) CDH can use the hydrolyzed products of cellulase or LPMO as a substrate, and can provide a sustainable supply of electrons without the addition of other substrates or light exposure; (2) CDH oxidizes the cellulose oligosaccharides and produces hydrogen peroxide at the same time which further promotes the efficiency of cellulose cleavage by LPMO (Figure 1) [26,27,28,29,30,31,32]; (3) Self oxidative deactivation rate of LPMO is lower when CDH is used an electron donor as compared to Asc [33].

Degradation of lignocellulose by cellulase requires a long period of time which is generally more than 24 h, and the optimum temperature for lignocellulose degradation by cellulase is usually around 45–50 °C [34]. Thus, the optimum temperature of auxiliary proteins such as CDH and LPMO should match the reaction conditions of cellulase to achieve the desired results. The optimum temperature of CDH from *Trametes hirsuta* (white-rot basidiomycete) is approximately 60 °C, however, it loses 70% activity at 50 °C even after 30 min [35]. In our study, we used a constitutive expression system for CDH gene expression of *Phanerochaete chrysosporium* in *Trichoderma reesei* which is termed as (*Pc*CDH). The expression product was purified to investigate its role as an electron donor of the AA9 family of LPMOs. The catalytic properties of *Pc*CDH were observed and we found that *Pc*CDH has an optimum temperature of 45 °C which is best suited for cellulase reaction. Furthermore, we observed that CDH while acting as an electron donor of LPMO, promotes the efficiency of cellulase during cellulose catalysis without the addition of extra substrate (cellobiose). Importantly, CDH as an electron donor has a very minor effect on the autoxidative inactivation of LPMO as compared to Asc.

## 2. Results

### 2.1. Heterologous Expression and Purification of PcCDH

The *Pc*CDH expression was carried out using the constitutive expression system of *Trichoderma reesei*, and we made the expression cassette using *T. reesei* promoter pdc1, CDH gene, and pdc1 terminator linked sequentially. The construct (Ppdc1-*Pc*CDH-Tpdc1) was co-transformed with pAN7-1 (which contains a hygromycin marker) into *T. reesei* for expression. The transformants were grown on PDA plates supplemented with hygromycin as a selection marker. The transformants were further confirmed through PCR and gel-electrophoresis. The selected transformants were incubated for 7 days on fermentation media which contains high contents of glucose. The total protein content was calculated through Bradford assay and the highest protein-producing transformant was selected for protein production and purification. The results of the SDS-PAGE analysis of proteins in the culture supernatant and purified proteins are shown in Figure 2. Two bands with molecular weights of approximately 100 and 80 kDa (not present in the control strain) appeared in the SDS-PAGE image, and western blot analysis showed that both bands contained a 6-histidine tag. Both bands were cut off and the proteins therein were identified with a mass spectrum. The results showed that both the bands were of *Pc*CDH because the peptides detected covered the whole *Pc*CDH protein. The theoretical weight of *Pc*CDH is approximately 82 kDa and we assume that the 100 kDa band is post-glycosylated *Pc*CDH, while 80 kDa is non-glycosylated *Pc*CDH. The results of liquid chromatography coupled with mass spectroscopy (LC-MS) confirmed that both the bands were *Pc*CDH. An important characteristic of expressing *Pc*CDH in *T. reesei* under the control of the pdc1 promoter is that the culture supernatant is very “clean”, that is, there are almost no other proteins expressed (Figure 2). We have tested the cellulase activity in the culture supernatant and found that there is no cellulase activity therein.

The Analysis of the absorption spectra for purified *Pc*CDH in both its reduced (blue) and oxidized (red) states, using UV/vis spectroscopy, revealed consistency with the typical CDH absorption spectra [36]. The main Soret band for the heme *b* cofactor was observed at 420 nm. Following the addition of cellobiose, the Soret band shifted to 430 nm, accompanied by the appearance of β and α peak maxima at 530 nm and 560 nm, respectively. The A_280_/A_420_ value for the purified recombinant *Pc*CDH is 0.29 (Figure 3), significantly lower than typically reported values (>0.6) [6,37]. This discrepancy may be attributed to the insufficient binding of the heme *b* cofactor within the cytochrome domain. The rapid synthesis of recombinant *Pc*CDH in *T. reesei* controlled by a strong *pdc* promoter contrasts with the potentially lower synthesis rate of heme *b. Despite only a proportion of recombinant PcCDH containing the* heme *b* cofactor, the kinetic parameters and the specific activity (11.57 U/mg) are comparable to previously reported values [6]. This equivalence is likely due to the DH domain’s ability to transfer electrons to external receptors, such as DCIP or LPMO [29].

### 2.2. Catalytic Properties of PcCDH

Kinetic constants for the oxidation of *Pc*CDH on different substrates at pH 4.0 with DCIP as the electron acceptor are listed in Table 1. The results showed that *Pc*CDH has the highest affinity for cellobiose relative to the other substrates and has a higher catalytic constant (k_cat_/K_m_) for cello-oligosaccharides than lactose. The optimum pH value for *Pc*CDH was 5.0 when cellobiose was used as the substrate and DCIP as an electron acceptor (Figure 4A). The optimum reaction temperature of *Pc*CDH was 50 °C (Figure 4B). There was no significant loss of activity of *Pc*CDH after 4 h of treatment in a metal bath at 40 °C, 1400 r/min. However, the *Pc*CDH activity remained only 20% at 50 °C, 1400 r/min in a metal bath after 4 h (Figure 4B).

### 2.3. Substrate Specificity of PcCDH

In addition, we analyzed the catalytic activity of *Pc*CDH for other substrates and found that *Pc*CDH has a short range of substrates with minimum substrate specificity. *Pc*CDH significantly catalyzed xylan birchwood and dextrin (Figure 5).

### 2.4. LPMO Activity Is Driven by CDH

The total volume of the LPMO lysis reaction system was 200 μL, containing 0.6 μM *Pc*CDH, 4.5 μM purified recombinant TtAA9F from *T. reesei,* 1 mM cellobiose, 33 mg/mL PASC. The lysis products were detected after 12 h of reaction by using an HRP chromogenic method based on gluco-oligosaccharide oxidase (Figure 6) [38].

As the cleavage products are oxidized by CDH to aldonic acid; thus, the amount of reducing sugars is very low at the end of the reaction and this method was found to be ineffective in the direct detection of reducing sugars. Therefore, we obtained the supernatant after the reaction by centrifugation. The supernatant was kept in boiling water for 5 min to inactivate the enzymes, cooled to room temperature, and then 2 μL of 1 IU/mL cellulase was added and the mixture was kept at 1400 r/min and 45 °C for 1 h. Then, we used the HRP method of Wu et al. and detected a significant increase in the amount of reducing sugars. Since cellulase causes further hydrolysis of oligosaccharide acid to short-chain reducing sugars; thus, we adopted this approach and followed the HRP method for the detection of LPMO lysis products in the subsequent assays (Figure 6). We found that *Pc*CDH acts as an efficient electron donor of LPMO while using PASC as a substrate and without the addition of cellobiose (Figure 6). *Pc*CDH gains electrons by oxidizing the reducing end of PASC and transferring them to the copper ion cofactor of LPMO.

### 2.5. Synergistic Effect with Cellulase

Cellulase-co-active proteins and their PASC hydrolysis efficiency were compared by the addition of different enzyme components. It is difficult to detect the product concentrations by DNS (3,5-dinitrosalicylic acid) assay in the presence of CDH. Therefore, we used a turbidimetric assay to study cellulase and LPMO’s ability to decompose PASC in the presence of *Pc*CDH [33]. The results are shown in Figure 7 and highlight the catalytic activity of LPMO against PASC in the presence of CDH. Very little synergistic effect was observed between LPMO and cellulase in the absence of an electron donor, while PASC degradation increased tremendously (up to 141%) in the presence of CDH at 12 h duration. Therefore, the desired combination involves the addition of CDH, cellulase, and LPMO to achieve efficient degradation of PASC.

### 2.6. PcCDH Delays LPMO Deactivation

Autoxidative deactivation of LPMO is a major concern during oxidative lysis of cellulose. Deactivation of LPMO is more rapid when Asc is used as an electron donor [39]. To compare the effect of CDH and Asc on LPMO deactivation, different reaction conditions were set. Three different reactions were set up which involve *Pc*CDH-LPMO, Asc-LPMO, and a control reaction containing LPMO but without an electron donor. The LPMO lysis activity was assayed, and the reaction conditions were set at 45 °C, and 1200 r/min at different time intervals. After 15 min, LPMO completely lost its activity while treated with Asc as an electron donor (Figure 8). However, LPMO retained 80% activity even after 10 h with CDH as an electron donor (Figure 8). There were on alterations in the LPMO activity even while using different concentrations of CDH, as assessed by comparison of different concentrations of CDH (Figure 8). Thus, we suggest that LPMO itself lost some enzymatic activity after being kept for 10 h at 45 °C (Figure 8). Thus, we believe that *Pc*CDH as an electron donor does not cause inactivation of LPMO.

### 2.7. MALDI-TOF-MS (Matrix-Assisted Laser Desorption/Ionization Time-of-Flight Mass Spectrometry) Analysis of LPMO Lysis Products

To further clarify the electron-donating ability of *Pc*CDH to LPMO, we analyzed the lysis products by MALDI-TOF-MS. Various degrees of polymerization (DP) of cello-oligosaccharides or their derivatives were detected in the presence of substrate with or without the addition of cellobiose (Figure 9). This further clarifies *Pc*CDH’s role as an electron donor of LPMO to drive its lysis activity regardless of the presence of cellobiose. The Peak map of the DP6 region in (Figure 9b) is enlarged and shown in (Figure 9c). The regions were consistent mainly on the following molecular ion peaks: peak 1009 corresponds to sodium adduct of LPMO double oxidized derivatives of natural hexasaccharide. Peak 1011 is consistent with the sodium adduct of the natural hexasaccharide lactone derivatives or sodium adduct of aldose derivatives. Peak 1027 confirms the potassium adduct of the natural hexasaccharide lactone derivatives. Peak 1029, corresponds to the potassium adduct of native hexasaccharide, the sodium adducts of the aldonic acid form of the hexasaccharid, or the sodium adduct of the gemdiol form of hexasaccharide. Peak 1049 corresponds to the dipotassium adduct of natural hexasaccharide lactone derivatives. Peak 1051 corresponds to the disodium adduct of the aldonic form of hexasaccharide.

## 3. Discussion

It is well documented that LPMO stimulates the cellulase activity for cellulose decomposition [40]. Cellulase plays a major role during this process and the auxiliary activity proteins have a supporting role in the cellulase-co-active protein system. Therefore, it is immensely important that the catalytic conditions of the component proteins should match accurately with the cellulase to achieve significant results. Thus, the use of CDH as an electron donor of the cellulase-LPMO system requires its screening based on optimum catalytic conditions. In this work, we investigated the electron donor effect of CDH of *P. chrysosporium* in *T. reesei*. for LPMO and its impact on the LPMO-cellulase catalytic activity during cellulose degradation. Our results have shown that *Pc*CDH precisely matches with the catalytic reaction conditions of cellulase in terms of temperature and pH and it serves as a potential electron donor of LPMO. This work confirmed that *Pc*CDH has optimum pH and temperature of 5 and 45 °C, respectively. Similarly, the optimum reaction temperature for cellulase is between 45–50 °C [41].

Therefore, we believe that *Pc*CDH is very well adapted to catalytic conditions of cellulase at medium temperature. Due to its high catalytic activity, *Pc*CDH acts as an efficient electron donor for the C1 type of LPMO. However, rapid deactivation of LPMO during catalysis is of great concern. Autoxidation of LPMO causes rapid deactivation, which may result due to the usage of Asc as an electron donor of LPMO [26]. We found that the addition of 1mM Asc results in complete deactivation of LPMO within 15 min. On the other hand, *Pc*CDH when used as an electron donor does not deactivate LPMO even while in the reaction for many hours. Hence, we can infer that *Pc*CDH is better for the longevity of LPMO activity, which improves the overall reaction through yield improvements and also proves cost-effective.

More than 80% of LPMO’s enzymatic activity was retained even after being held for 10 h with the addition of *Pc*CDH. The reaction was carried out in the absence of substrate. In another reaction we found that LPMO still loses its activity in the absence of an electron donor, thus the presence of *Pc*CDH may preserve the catalytic activity of LPMO. Therefore, we believe that the presence of *Pc*CDH as an electron donor of LPMO does not cause its inactivation.

Moreover, the degradation of PASC by cellulase was improved significantly with the addition of *Pc*CDH alongside LPMO. We suggest two reasons for this phenomenon, (1) *Pc*CDH serves as an electron donor of LPMO which results in oxidative lysis of crystalline cellulose and renders cellulose more susceptible to cellulase hydrolysis [42]; (2) *Pc*CDH may alleviate product inhibition on cellulase-caused by cellobiose [43]. *Pc*CDH intensifies the oxidative lysis of cellulase products to glycolic acid, which reduces cellulase inhibition by oligosaccharides such as cellobiose [44]. Generally, a small addition of lactose or cellobiose is necessary to drive the LPMO activity while CDH is being used as an electron donor [29,33]. However, we found that *Pc*CDH does not require the addition of any substrate to drive the activity of C1-type LPMO on PASC. This proves the strong electron donor ability of *Pc*CDH to LPMO during the oxidation of PASC, and the lysis products of PASC by C1-type LPMO are in turn oxidized by *Pc*CDH. The oligosaccharide products produced by cellulase-LPMO degradation system can also act as substrates for *Pc*CDH, thus *Pc*CDH does not require the addition of cellobiose and can still effectively transfer electrons to LPMO.

## 4. Methods and Materials

### 4.1. Cultivation Conditions, Reagents, Plasmids and Strains

We used DH5α (*Escherichia coli*) for plasmid construction. For the propagation and cultivation of bacterial strains, we used LB medium (1% NaCl, 0.5% yeast extract, and 1% tryptone supplemented with ampicillin 100 μg/mL if necessary). We used *Trichoderma reesei* QM9414 (ATCC 26921) strain for the heterologous protein expression. *T. reesei* cultivation conditions were followed from [45]. PDA agar was prepared by the addition of potato infusion 20%, dextrose 2%, and 2% agar and used for the cultivation of *Thielavia terrestris* (ATCC 38088) at 45 °C. Expression cassettes were constructed using plasmid pUC19. An assisting plasmid pAN7-1 (hygromycin B containing plasmid) was used for the transformation process of *T. reesei* [46].

PASC (phosphoric acid swollen cellulose) was prepared from Avicel PH-101 (Sigma-Aldrich, St. Louis, MO, USA) following the protocol of Canella et al. [47]. 2,6-Dichlorophenolindophenol (DCIP) was used for enzyme activity, temperature, and pH optima and was purchased from Sangon (Shanghai, China). Ascorbate oxidase was purchased from Solarbio (Beijing, China), and the chemicals of analytical grade were bought from BBI Life Sciences Corporation (Shanghai, China).

### 4.2. Expression Vector Construction & Transformation to Fungal Protoplast

The genomic DNA of *T. terrestris* and *T. reesei* was extracted with a DNA extraction kit (Sangon Biotech, Shanghai, China). The promoter and terminator sequence of *pdc1* and *cdna1* genes were PCR amplified from *T. reesei* QM9414 genomic DNA (Table 1 representing the primer sequence). Similarly, the TtAA9F gene (GenBank sequence number: THITE-2142696) and its native signal peptide (SP) sequence were PCR amplified from *T. terrestris* genomic DNA. During PCR amplification 6×histidine tag was added at each gene’s C-terminus. The sequential ligation of P*pdc1*, respective gene, and T*pdc1* into a pUC-19 vector and achieved by ClonExpress II One Step Cloning Kit for the construction of expression cassette (Vazyme Biotech, Nanjing, China). The linearized constructs of pUC-19 and pAN7-1 were co-transformed in *T. reesei* protoplast by the polyethylene glycol method followed from [46].

CDH gene sequence was obtained from *Phanerochaete chrysosporium* strain OGC101 from GenBank (entry U46081.1) which was commercially synthesized after codon optimization by IGEbio (Guangzhou, China), with the addition of the signal peptide sequence of *T. reesei cbh1* at the N-terminus and a 6×histidine tag at the C-terminus for expression in *T. reesei*. Construction of expression cassettes was achieved by the sequential ligation of the *pdc1* promoter, *PcCDH* gene, and the *pdc1* terminator into pUC19 plasmid with a ClonExpress II One Step Cloning Kit (Vazyme Biotech, Nanjing, China). The expression cassette was expressed in *T. reesei* QM9414 protoplast to achieve heterologous expression of CDH under the *pdc1* promoter. The linearized pAN7-1 (contains selection marker hygromycin B) and pUC19 were co-transformed in *T. reesei* QM9414 protoplast [46].

Finally, different amounts of protoplast (100, 200, 300 μL) containing the expression vector and the selectable vector were suspended on PDA supplemented with 100 μg/mL hygromycin B. The plates were kept at 28 °C for 36–48 h and these colonies were transferred to PDA plates supplemented with 50 μg/mL hygromycin B. The colonies were confirmed through PCR (see Table 2 for primers used in this study) using One *Taq* DNA Polymerase from Vazyme (Nanjing, China) following the user manual, and band visualization through gel electrophoresis.

### 4.3. Production and Purification of Recombinant Protein

For the production of recombinant protein in *T. reesei* we followed Li et al. [45]. The total protein concentration was determined by a standard Bradford reagent kit (Sangon Biotech, Shanghai, China). Protein size was determined by Western blot analysis using HRP-conjugated rabbit anti-mouse IgG and anti-6 × His tag mouse monoclonal antibody (BBI Life Sciences Corporation, Shanghai, China). ÄKTA Purifier UPC100 FPLC-System (GE Healthcare) was used to purify the recombinant proteins as follows. First, we filtered the supernatant with Whatman filter paper to remove insoluble residues. This filtrate was then filtered with a polyvinylidene fluoride (PVDF) membrane of 0.45 μm pore size. Finally, this filtrate was further filtered with a PVDF membrane of 0.2 μm pore size. This double-filtered filtrate was kept in a dialysis bag with 7000 Da molecular weight cut off and placed in a tray. PEG 20,000 powder was sprinkled over the dialysis bag covered completely with this powder and incubated for 12 h at 4 °C to obtain higher protein concentration through osmosis. This dialysis bag having the concentrated protein sample was again dialyzed using the affinity chromatography binding buffer (sodium phosphate 20 mM, NaCl 500 mM, pH 8.0) and kept for 12 h, and in this duration, the buffer was changed once.

Ni Sepharose 6 Fast Flow on the FPLC system was used to purify the recombinant protein. Eluent A (sodium phosphate 20 mM, NaCl 500 mM, pH 8.0) & eluent B (eluent A + 500 mM imidazole) flow rates were adjusted to fulfill the gradient elution condition, which means that imidazole concentration was maintained at 25 mM in the loading stage, while in the washing stage its concentration was adjusted in a stepwise gradient (50–75 mM) to remove any proteins which are nonspecifically bound. To elute the target protein, the imidazole concentration in the elution buffer was maintained at 250 mM. The purified protein fractions were concentrated by Amicon Ultra Centrifugal Filters (Millipore, Germany) and purified further by gel filtration chromatography through a Superdex 200 Increase 10/300 GL column (GE Healthcare) and eluted with 150 mM NaCl and 20 mM Tris (pH 7.5) at 0.5 mL/min flow rate. Till this point, the protein production and purification for the recombinant *Pc*CDH and LPMO proteins are the same.

However, an equal volume of copper solution (at a concentration of 3 times the LPMOs) was mixed with purified LPMO solution to recover the Cu^2+^ loss during purification; and incubated at 4 °C for 1 h. Residual copper was removed by desalting through Sephadex G-25 Medium (GE Healthcare, Chicago, IL, USA).

### 4.4. MS Analysis of Purified Recombinant PcCDH

For the mass spectrum analysis of the recombinant *Pc*CDH, gel pieces containing the target protein bands were cut, de-stained, dehydrated, and digested with trypsin. After digestion, peptides were extracted and processed for matrix-assisted laser desorption ionization-time of flight (MALDI TOF/TOF) MS analysis (ABI 4800 MALDI-TOF/TOF proteomics analyzer mass spectrometer, AB SCIEX).

### 4.5. Characterization of Catalytic Activity of PcCDH

*Pc*CDH activity analysis was performed by the following method. The total volume of the reaction mixture was 200 μL, consisting of purified *PcCDH* (0.05 μM), sodium acetate buffer 50 mM (pH 4.0), cellobiose (500 μM) and DCIP (0–3 μM). The reaction was carried out at room temperature and the OD was measured dynamically at 520 nm using a Synergy MX enzyme calibrator. The OD value of the 60s was selected for the calculation of the initial speed of the enzyme reaction and activity. One unit is defined as the amount of enzyme needed to reduce 1 μmol DCIP per minute under the selected assay condition (pH 4.0, 25 °C) [48].

Kinetic constants of *Pc*CDH were determined as follows. The reaction mixture contained 0.3 mM DCIP, 0.05 μM *Pc*CDH, 50 mM sodium acetate buffer (pH 4.0), and different substrates, and the OD value was measured dynamically at 520 nm using a Synergy MX enzyme calibrator. Different substrates were used, including lactose (0–640 μM), cellobiose (0–640 μM), cellotriose (0–140 μM), cellotetraose (0–140 μM), cellopentaose (0–168 μM) and cellohexose (0–140 μM).

To analyze the effect of pH on enzyme activity, Britton-Robison buffer at pH 2.0–11.0 was used to adjust the pH. The reaction system contained 100 mM BR buffer, 500 μM cellobiose, 0.3 mM DCIP, 0.3 μM *Pc*CDH. The enzymes were kept separately under the different pH buffers for 3 h at 4 °C, the substrate was added with DCIP, and the reaction rate was calculated by measuring OD_520_ at the 60 s of the reaction.

To analyze the temperature effect on enzyme activity, a PCR instrument was used to adjust the temperature from 20 to 80 °C. The reaction mixture was comprised of 100 mM phosphate buffer (pH 7.0), 500 μM cellobiose, and 0.3 μM *Pc*CDH. After the PCR instrument was left for 10 min, 50 μM Amplex Red with 2 μL of 40 U/mL HRP was added and OD_560_ was measured to calculate the reaction rate. To analyze the temperature effect on the stability of the enzyme, the enzyme was kept at different temperature ranges of 40–70 °C for 4 h, and then 50 μM Amplex Red with 2 μL of 40 U/mL HRP was added and finally, OD_560_ was measured to calculate the reaction rate. The catalytic activities of *Pc*CDH on other substrates were analyzed, as the reaction mixture contained 100 mM phosphate buffer (pH 7.0), 50 μM Amplex Red, 2 μL of 40 U/mL HRP, 0.3 μM *Pc*CDH, and different substrates. Other substrates used for the catalytic activities were cellobiose (2 mM), D-glucose (2 mM), Maltose (2 mM), Xylan birchwood (2 mM), a-Maltose (2 mM), Dextrin (2 mM), Xylobiose (2 mM), Xylotriose (2 mM), Pectin (0.1 mg/mL), Trehalose dihydrate (0.1 mg/mL).

### 4.6. LPMO Cracking Efficiency Analysis

The reaction system contained phosphate buffer 100 mM (pH 7.0), PASC 33 mg/mL, LPMO 4.5 μM, *Pc*CDH 0.6 μM, and cellobiose or cellotriose 1 mM. Experiments were also carried out without the addition of cellobiose or cellotriose. The reaction was set at 1200 r/min and 50 °C (metal bath, 5355, Eppendorf) for 12 h. At the end of the reaction, a set of experiments was kept in a 100 °C water bath to inactivate the enzyme for 5 min and the supernatant was taken for analysis of the lysis product after cooling (no cellulase hydrolysis was added and the product was analyzed directly). Lysis products were detected followed by Wu & colleagues [38]. In another set of experiments, the supernatant was inactivated at 100 °C for 5 min in a water bath and then cooled, later kept in a new 1.5 mL centrifuge tube, added 2 μL of cellulase (1 IU/mL), reacted at 50 °C for 1 h at 1200 r/min, inactivated again at 100 °C for 5 min in a water bath, then cooled, and the supernatant was centrifuged for lysis product analysis (lysis product with cellulase was analyzed after hydrolysis of the LPMO).

### 4.7. MALDI-TOF-MS Analysis

After centrifuging the lysis products were combined with the substrates 5-chloro-2-mercapto-benzothiazole (CMBT) and 2,5-dihydroxybenzoic acid (DHB) to form co-crystalline films on the surface of the sample target and the products were analyzed using 5800 MALDI-TOF-MS (AB SCIEX 5800) as described by Chen et al. [49]. The MS data acquisition mass range was from 500 to 2500 *m*/*z*.

### 4.8. Electron Donor Effect on LPMO Deactivation

The experimental scheme was divided into two stages: one is the pretreatment assay for lysis activity. The pretreatment phase was carried out in 1.5 mL centrifuge tubes containing 9 μM LPMO and 50 mM phosphate buffer (7.0); 1 mM Asc; 1.2 μM *Pc*CDH, consecutively added in it for a total volume of 200 μL, with 500 μM cellobiose added to the tube containing *Pc*CDH. The pretreatment was carried out by incubating the above mixture at 50 °C, 1200 r/min in a metal bath at different times (0, 5 min, 10 min, 15 min, 30 min, 1 h, 3 h, 5 h, 10 h). The lysis activity was assayed after the pretreatment. The total reaction mixture consisted of 200 μL, including 100 μL of pretreatment sample, PASC (33 mg/mL), 0.5 mM Asc or 0.3 μM *Pc*CDH, 50 °C, 1200 r/min for 12 h. After inactivation in a water bath at 100 °C for 5 min, the supernatant was centrifuged and assayed for LPMO residual enzyme activity. The different concentrations (0, 0.1, 0.2, 0.4, 0.8, 1.6, and 3.2 μM) of *Pc*CDH were used to test the effect on LPMO activity.

### 4.9. Synergistic Effect of CDH, Cellulase and LPMO on Cellulose Deconstruction

Reactions were performed in 24-well plates containing 1.5 IU/mL cellulase, 3 μM LPMO, 0.35 μM *Pc*CDH, phosphate buffer (pH 7.0), and PASC (33 mg/mL) in a total volume of 1 mL. The other experimental groups were cellulase; cellulase and LPMO; cellulase and *Pc*CDH; *Pc*CDH and LPMO; LPMO at 50 °C, 200 r/min for 12 h, 24 h, and 48 h. The OD values were measured at 620 nm using a Synergy MX enzyme marker, following the methods described by Filandr et al. [33]. The remaining concentration of PASC after the reaction was calculated from the PASC concentration-OD620 standard curve and the relative hydrolytic activity was calculated from the rate of consumption of the PASC substrate.

## Figures and Tables

**Figure 1 ijms-24-17202-f001:**
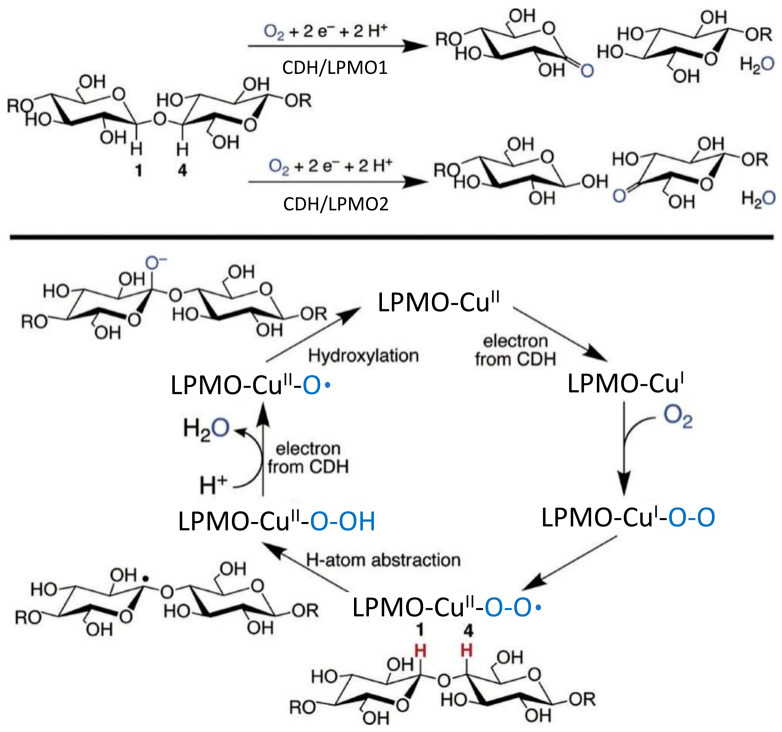
Proposed reaction mechanisms of CDH as an electron donor of LPMO (**Top**) represents Type 1 or C1-LPMOs extracts H atom from position 1 and leads to lactone sugars formation. However, the Type 2 or C4-LPMOs break the H atom extracted from position 4 and lead to ketoaldoses formation. (**Bottom**) represents the electron donation from CDH which reduces Cu (II) of LPMO to Cu(I) and then binds to O_2_. Copper superoxo intermediate results due to this internal transfer of electron and it can abstract the hydrogen ion from position 1 or 4 of the carbohydrate. Cu-bound hydroperoxide homolytic cleavage results due to a second electron from CDH. The substrate hydroxylation occurs when the substrate radical couples with copper oxo species (Cu−O•). The glycosidic bond is destabilized due to the addition of an oxygen atom and adjacent glucans are eliminated [8].

**Figure 2 ijms-24-17202-f002:**
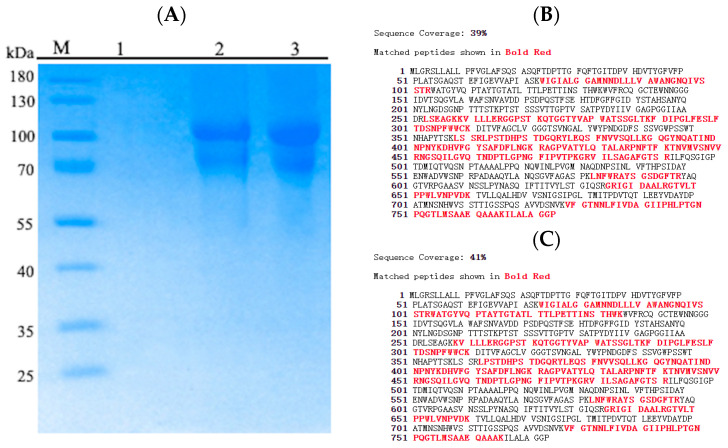
SDS-PAGE analysis of recombinant *Pc*CDH protein. (**A**) SDS-PAGE analysis of *Pc*CDH. M: molecular marker, 1: culture supernatant of recombinant *Pc*CDH, 2: purified recombinant *Pc*CDH, 3: repeat of 2. (**B**) Identification of the 80 kDa band in the *Pc*CDH electrophoresis photograph; and (**C**) Identification of the 100 kDa band in the *Pc*CDH electrophoresis photograph. The red peptides are identified using MS through molecular weight match.

**Figure 3 ijms-24-17202-f003:**
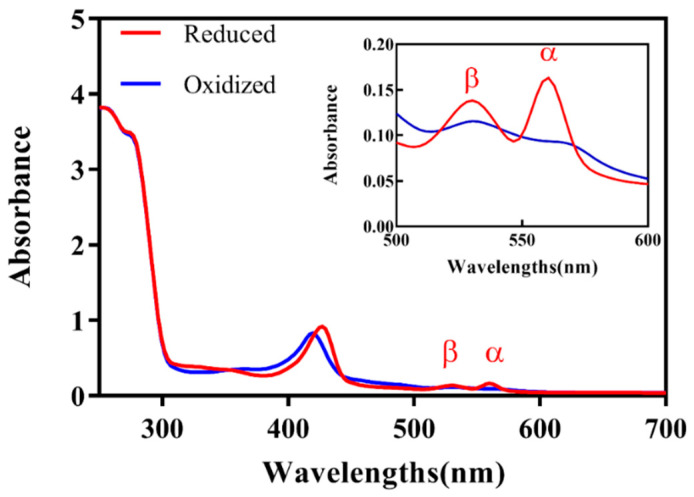
In spectral analysis of *Pc*CDH, the blue line represents oxidized while the red line represents reduced UV/Vis spectra of *Pc*CDH. The inset shows the Magnified spectral range (500–600 nm) of heme *b* α- and β-peak.

**Figure 4 ijms-24-17202-f004:**
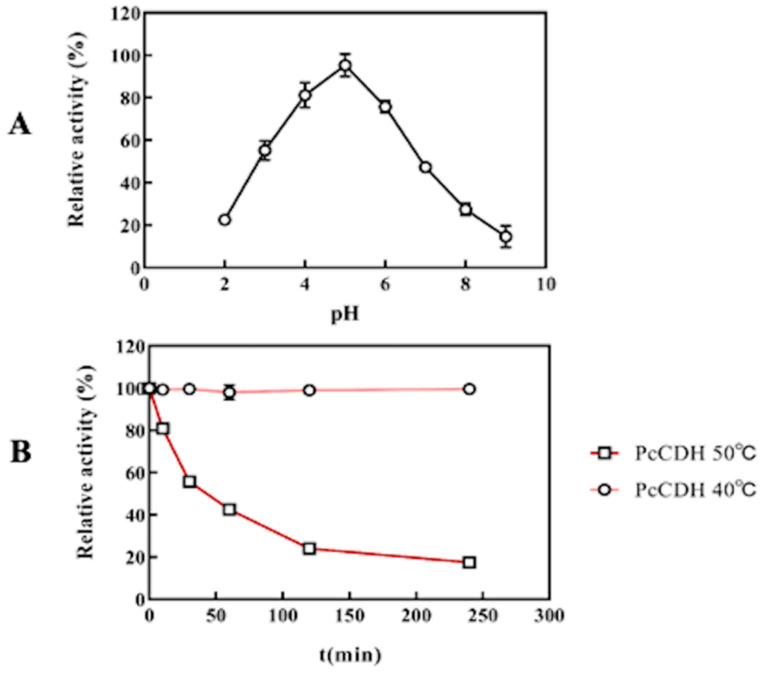
Basic catalytic properties of *Pc*CDH (**A**) Optimum pH of *Pc*CDH. (**B**) Optimum reaction temperature and thermal stability of *Pc*CDH at 40 °C and 50 °C. To analyze the effect of pH on enzyme activity, Britton-Robison buffer at pH 2.0–11.0 was used to adjust the pH. The reaction system contained 100 mM BR buffer, 500 μM cellobiose, 0.3 mM DCIP, 0.3 μM *Pc*CDH. Error bars indicate the standard deviation (*n* = 3; independent experiments).

**Figure 5 ijms-24-17202-f005:**
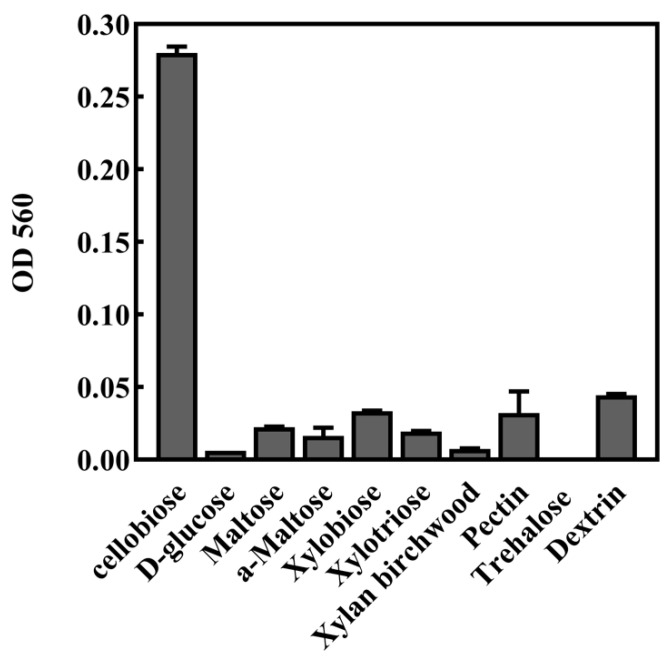
Substrate specificity of *Pc*CDH. The catalytic activity of *Pc*CDH on different substrates was analyzed through the Amplex red-horseradish method to measure the H_2_O_2_ production. The reaction mixture contained 100 mM phosphate buffer (pH 7.0), 50 μM Amplex Red, 2 μL of 40 U/mL HRP, 0.3 μM *Pc*CDH and different substrates. Error bars indicate the standard deviation from three independent experiments.

**Figure 6 ijms-24-17202-f006:**
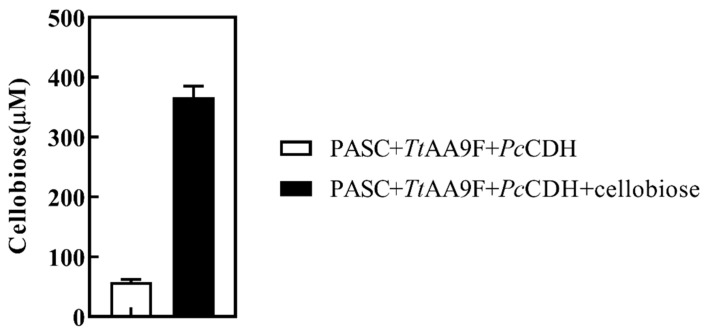
*Pc*CDH-LPMO lysis products. The blank bar shows the detection of PASC lysis products using LPMO and CDH. The black-colored bar in the graph shows the number of lysis products of PASC using LPMO, CDH, and cellobiose. Lysis products were assayed by the addition of extra cellobiose or cello-triose in the presence of PASC as substrate. Error bars indicate the standard deviation from three independent experiments.

**Figure 7 ijms-24-17202-f007:**
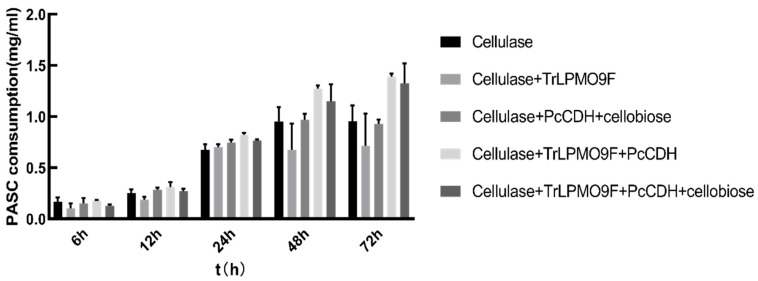
Synergistic effect of *Pc*CDH, LPMO, and cellulase. PASC degradation by cellulase alone or in different combinations with LPMO and *Pc*CDH (cellulase, cellulase + LPMO, cellulase + *Pc*CDH + cellobiose, and cellulase + LPMO + *Pc*CDH + cellobiose) was measured through turbidimetry method. Initially, the effect of cellulase alone or in combinations had no obvious difference, however, with time cellulase alone or its combination (cellulase + LPMO + *Pc*CDH) produced the best results at 12-24 h. Afterward, cellulase showed no further increment but its combination with LPMO and *Pc*CDH still resulted in maximum degradation of PASC at 72 h. Error bars indicate the standard deviation (n = 3; independent experiments).

**Figure 8 ijms-24-17202-f008:**
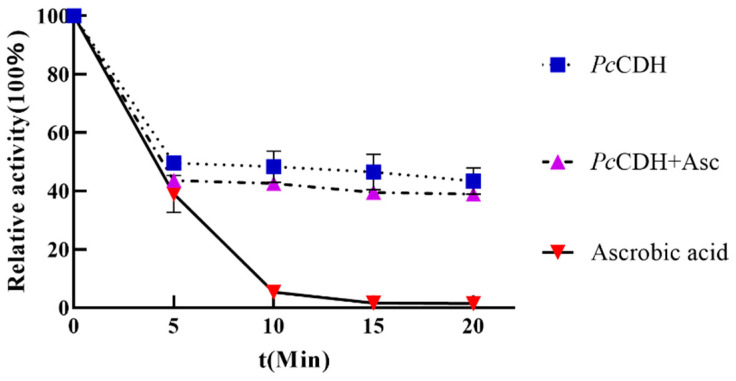
LPMO activity and its deactivation under various electron donors. *Pc*CDH results in delayed autoxidative deactivation of LPMO and prolongs its enzymatic activity as compared to Asc when it is used alone. However, Asc when used alone or in combination with *Pc*CDH increases the oxidative deactivation of LPMO.

**Figure 9 ijms-24-17202-f009:**
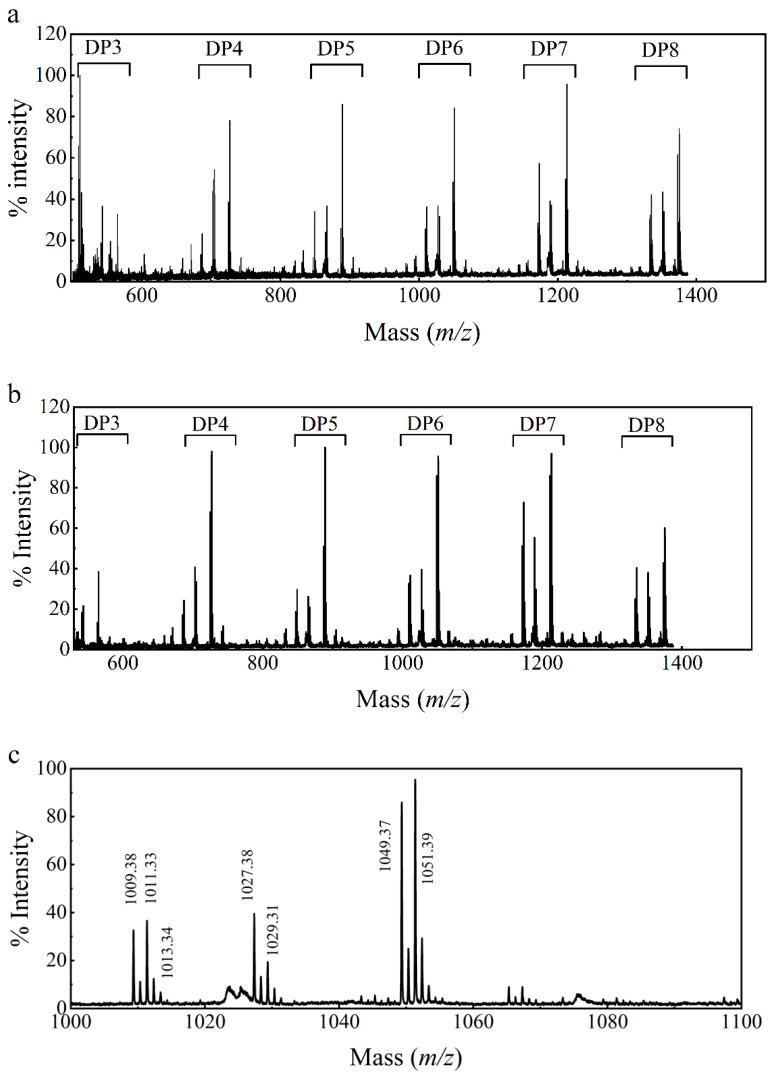
LPMO and *Pc*CDH product analysis through MALDI-TOF-MS (**a**) Products produced by LPMO when the substrate is PASC. (**b**) Product produced by LPMO when the substrate is PASC with cellobiose. (**c**) Magnified DP6 peaks of (**b**).

**Table 1 ijms-24-17202-t001:** Kinetic constants of *Pc*CDH.

Electron Donors	*PcCDH*		
	K_m_	k_cat_	k_cat_/K_m_
	[mM]	[s^−1^]	[mM^−1^ s^−1^]
Lactose	0.97 ± 0.17	30.94 ± 3.70	31.77
cellobiose	0.10 ± 0.01	21.66 ± 0.78	206.34
cellotriose	0.36 ± 0.06	52.76 ± 7.51	144.17
cellotetraose	0.16 ± 0.03	21.57 ± 2.67	134.01
cellopentaose	0.35 ± 0.09	47.53 ± 8.95	133.90
cellohexose	0.30 ± 0.06	27.17 ± 4.65	89.96

**Table 2 ijms-24-17202-t002:** Primers used in this study.

Primer	Sequence	Application
Pcdna1-F	aaaacgacggccagtgaattcCAGACAATGATGGTAGCAGCGC	Promoter amplification
Pcdna1-R	gcttcgaccgagcatTTGAGAGAAGTTGTTGGATTGATCA	Promoter amplification
CDH-F	ctcaaATGCTCGGTCGAAGCCTCCTCGC	Gene Amplification
CDH-R	GGttaatgatgatgatgatgatgGGGTCCTCCGGCGAGAGC	Gene Amplification
Tpdc1-F	CcatcatcatcatcatcattaaCCCGGCATGAAGTCTGACC	Terminator Amplification
Tpdc1-R	gattacgccaagcttgcatgcTGGACGCCTCGATGTCTTCC	Terminator Amplification

## Data Availability

Data are contained within the article.

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
