# Peer review of "Heterologously Expressed Cellobiose Dehydrogenase Acts as Efficient Electron-Donor of Lytic Polysaccharide Monooxygenase for Cellulose Degradation in *Trichoderma reesei"

_ijms, 2023, doi:10.3390/ijms242417202_

Round 1

Reviewer 1 Report

Comments and Suggestions for Authors

This study shows that CDH of Phanerochaete chrysosporium synergizes with cellulase preparations of Trichoderma reesei to improve cellulose degradation efficiency. the synergistic effect of CDH and cellulase has already been reported (in reference 42, 43). Therefore, the results on synergism are not new, but the finding that CDH acts as an electron donor for C1 type LPMO is valuable.

     A concern in this study is that CDH is heterologously expressed in T. reesei. Despite the glucose culture, cellulase is produced with extremely little amounts. Therefore, evidence of the absence of cellulase contamination is needed.

     The meaning of using both the pdc promoter and the cDNA1 promoter is unclear. If the expression potency of the two promoters is to be compared, it should be clearly stated.

Author Response

We are thankful to your vaulable opinions and made changes in the manuscript according to your suggestions. 

This study shows that CDH of Phanerochaete chrysosporium synergizes with cellulase preparations of Trichoderma reesei to improve cellulose degradation efficiency. the synergistic effect of CDH and cellulase has already been reported (in reference 42, 43). Therefore, the results on synergism are not new, but the finding that CDH acts as an electron donor for C1 type LPMO is valuable.

Question 1:

A concern in this study is that CDH is heterologously expressed in T. reesei. Despite the glucose culture, cellulase is produced with extremely little amounts. Therefore, evidence of the absence of cellulase contamination is needed.

Response: Yes, it is really a concern that T. reesei might produce cellulase even in culture medium with high glucose concentration. However, we have analyzed the cellulase activity in the culture supernatant and found that there isn’t any cellulase activity within 7 days of cultivation. As, all the activity data is nearly zero, therefore, we haven’t provided a figure. Thus, we have just described the result in our revised manuscript.

 Question 2:

The meaning of using both the pdc promoter and the cDNA1 promoter is unclear. If the expression potency of the two promoters is to be compared, it should be clearly stated.

Response: We have used both pdc and cDNA1 promoters for PcCDH expression, and found that pdc promoter is more effective. As, the emphasis of this manuscript is to study the enzymatic characteristics of the recombinant PcCDH and its potential to act as an electron donor of LPMO, therefore, we have removed the contents related to cDNA1 promoter in the revised manuscript.

Reviewer 2 Report

Comments and Suggestions for Authors

In the research manuscript submitted by Muhammad Adnan et al., the authors produce cellobiose dehydrogenase (CDH) from P. chrysosporium in T. reesei and provide a detailed characterization of the enzyme's properties. Additionally, they investigate the synergistic action of the obtained CDH with LPMOs for the deconstruction of PASC. While the authors present a substantial amount of data, several concerns need addressing, and certain sections of the manuscript require careful revision and language editing. Firstly, I need to mention that the expression and characterization of the same enzyme in T. reesei have been reported previously, which may limit the novelty of the current submission. CDH from P. chrysosporium has also already been characterized in various studies before. Some experiments, such as the verification of protein purity, appear to be missing from the manuscript. It is recommended that these experiments be included, and the authors should explicitly address whether their enzyme preparation is homogeneous or if it consists of a mixture of the isolated DH domain and the full-length CDH. This issue is particularly relevant to Figure 3, and I have provided additional comments on this aspect.

I hope that the authors find these critiques constructive in refining and enhancing the quality of their manuscript:

Figure 1: The authors present a homology model of PcCDH created by Swissmodel. What does the color schema (red, blue, white) show? Surface charge? Confidence score? Furthermore, the heme b cofactor of the cytochrome domain is missing while the FAD of the dehydrogenase domain is in. The authors need to show both cofactors. Why do the authors use a homology model at all? The AlphaFold model of PcCDH is available on uniport. Also, the resolved crystal structures of the DH (pdb ID: 1KDG) as well as pf the CYT domain (pdb ID: 1D7C) are available.

Figure 2: The authors just present the oxygen-dependent monooxygenase reaction of LPMO. However, especially for fungal AA9 LPMOs it is known the H2O2-dependent peroxygenase reaction is orders of magnitudes faster and the predominant reaction also under natural conditions. (DOI: 10.1038/nchembio.2470, https://doi.org/10.1038/s41467-022-33963-w) The authors shortly mention the importance of H2O2 briefly in their text (Line 87)

Figure 3: What is the difference between the samples in lane 2 and 3 or lane 4 and 5, respectively? Is the protein expressed under the control of different promoters applied twice? Please also correct “KDa” to “kDa”.

The authors should show an SDS-PAGE result to confirm the purity of their protein, not only a western blot. Just from the Western blot it is not possible to prove if the protein is pure or not.

Line 131f.: The authors found 2 distinct bands of their target protein by Western blot analysis. The argue that this is due to different glycoforms. However, in my opinion the bands are to sharp to account for two distinct glycoforms. Form other studies it is well known that CDH can be proteolytically cleaved into the DH and the CYT domain. The isolated DH domain is often found as when purifying CDHs from native fungi as well as expression hosts. In a previous paper PcCDH was also heterologously expressed in T. reesei and the authors reported to obtain the DH domain as well as the full-length CDH. (https://doi.org/10.1186/s12934-020-01492-0). The size of the lower band corresponds very well to the size of the DH domain and the results presented by the authors here are very similar to the observations made in the paper mentioned before. The authors need definitely to consider this and check the possibility of the presence of the DH domain.

Figure 4: Which assay was used for the pH-optimum? DCIP as electron accepter? Cellobiose as electron donor? Which concnetrations? Please provide this information in the Figure’s caption.

Figure 5: Why is the absorbance of the protein so high at 280 nm compared to the absorbance of the heme at 420 nm? Native PcCDH has an ratio of A420/A280 of > 0.6 (https://doi.org/10.1016/S0167-4838(97)00180-5) and the same is true for heterologously produced PcCDH (https://doi.org/10.1186/s12934-020-01492-0). The spectrum reported here shows an A420/A280 of < 0.3 which would mean that the protein purity is very low or that the heme loading is insufficient. The authors definitely need to comment on that.

Figure 6: What do the authors measure at 560 nm? Please explain in the figure caption. Furthermore, the authors need to check the purity of their preparations (especially xylan and gum Arabic) as the KM for cellobiose and other oligosaccharides is very low, already small contents of these sugars would lead to false positive results.

Figure 7: How much cellobiose was added in the second sample? I cannot find 7a and 7b here but only one plot with two samples. Please check and correct description.

Figure 8: Also, here the Figure caption needs improvement. Which assay was used? Turbidimetry?

The time scale between the experiments presented in Figure 8 (72 h) and Figure 9 (15 min) are completely different. The authors definitely need to explain better why the see the boosting effect of LPMO only after more then 24 h while inactivation of LPMO seems to occur already much faster (min – h scale)

Minors

Line 27f.: This sentence is not clear to me. Is there a mistake in the sentence?

Line 44f. I think it would be too generic to state that the specific function of oxidative enzymes of fungi is “elusive” There are a lot of studies targeting this topic and reasonable models for the function of many oxidoreductases in lignocellulose deconstruction.

Line 53: Do the authors mean longer cellooligosaccharides instead of “longer cellodextrin”?

Line 58: Grammar and style needs to be improved

Line 62: Quinones can also be reduced at the isolated DH domain (e.g. 1,4-benzoquinone, 2,6- dimethyl-1,4-BQ, etc.). The cytochrome domain is not involved in their reduction. The same is true for DCIP while cytochrome c can only be reduced via the cytochrome domain.

Line 64: Replace “Cytochrome C” by “cytochrome c”

Line 67: Regarding LPMO classification the authors should also include recent literature and the addition of another family of LPMOs (AA17).

Line 68: Grammar needs to be corrected (…belongs…) and language and style need to be improved.

Line 69. The authors write: “These LPMOs in synergism with cellulases can oxidatively cleave crystalline cellulose through hydrolysis”. This is misleading as LPMO is not performing hydrolysis but oxidative cleavage of the carbohydrate backbone.

Line 80f.: The authors use the “terms inter-domain electron transfer” (IET), “direct electron transfer (DET)” and “electron transfer” not correctly. IET is defined as the electron transfer form the DH to the CYT domain and not to an “external electron acceptor”. DET is defined as the electron transfer from the enzyme to an electrode without mediator.

Line 96: grammar (loses instead of losses).

Line 127f. The authors just measured the total protein concentration to identify the best producer. What is the proportion of CDH relative to the remaining proteins in the culture supernatant? Why are the authors not using a CDH-specific activity assay (DCIP, etc.)?

Line 152: change “kinetic parameters” to “kinetic constants”

Line 154f.: The authors write: “PcCDH has a lower affinity for the substrate (lower Km), more catalytic for the bound substrate (higher kcat). Similarly, PcCDH has a higher catalytic constant (kcat/Km) for fibrous oligosaccharides.” – This is absolutely not clear for me and I think needs proper revision. Furthermore, what do the authors mean by “fibrous oligosaccharides”? Cellooligosaccharides?

Table 1: correct “Kcat” to “kcat”.

Table1: The authors use to many significant digits (especially for kcat values) please check and apply correct rounding of measurement results.

Line 359: What is the ascorbate oxidase used for in this study?

Line 385, Line 388: Please chang uL to “µL” and “ug/mL” to “µg/mL”

Line 397: correct “ÄKTA”

Line 480: What do the authors mean by cellulose disaccharides or cellulose trischaccharides?

There needs to be space between number and unit. Please correct. E.g. Line 318, Line 469, Line 474, Line 475, Line 476, Line 448, etc.

Comments on the Quality of English Language

Author Response

We really appreciate your time and effort in critically analyzing our manuscript, without which we may have ben unable to improve our article. We have tried our best to address each and every comment raised by you. Following are our responses. 

Reviewer 2:

In the research manuscript submitted by Muhammad Adnan et al., the authors produce cellobiose dehydrogenase (CDH) from P. chrysosporium in T. reesei and provide a detailed characterization of the enzyme's properties. Additionally, they investigate the synergistic action of the obtained CDH with LPMOs for the deconstruction of PASC. While the authors present a substantial amount of data, several concerns need addressing, and certain sections of the manuscript require careful revision and language editing.

Question 1:

Firstly, I need to mention that the expression and characterization of the same enzyme in T. reesei have been reported previously, which may limit the novelty of the current submission. CDH from P. chrysosporium has also already been characterized in various studies before.

Response: Yes, PcCDH has been produced in T. reesei and its enzymatic characteristics have been studied before. However, in our study, we have found that PcCDH when used as an electron donor of LPMO, it can work well without the addition of cello-oligosaccharides such as cellobiose, and the auto-oxidative deactivation of LPMO is alleviated.

Question 2:

Some experiments, such as the verification of protein purity, appear to be missing from the manuscript. It is recommended that these experiments be included, and the authors should explicitly address whether their enzyme preparation is homogeneous or if it consists of a mixture of the isolated DH domain and the full-length CDH. This issue is particularly relevant to Figure 3, and I have provided additional comments on this aspect.

Response: We apologize for the mistake. Actually, we have purified the recombinant PcCDH. However, we haven’t shown the purification results in the first version of the manuscript by mistake. We have replaced the figure in revised manuscript. The two bands are full length PcCDH as identified by mass spectrum analysis, and the results are shown in the revised manuscript.

I hope that the authors find these critiques constructive in refining and enhancing the quality of their manuscript:

Figure 1: The authors present a homology model of PcCDH created by Swissmodel. What does the color schema (red, blue, white) show? Surface charge? Confidence score? Furthermore, the heme b cofactor of the cytochrome domain is missing while the FAD of the dehydrogenase domain is in. The authors need to show both cofactors. Why do the authors use a homology model at all? The AlphaFold model of PcCDH is available on uniport. Also, the resolved crystal structures of the DH (pdb ID: 1KDG) as well as of the CYT domain (pdb ID: 1D7C) are available.

Response: As the structural model of PcCDH can be easily obtained, thus we have deleted this figure in the revised manuscript.

Figure 2: The authors just present the oxygen-dependent monooxygenase reaction of LPMO. However, especially for fungal AA9 LPMOs it is known the H2O2-dependent peroxygenase reaction is orders of magnitudes faster and the predominant reaction also under natural conditions. (DOI: 10.1038/nchembio.2470, https://doi.org/10.1038/s41467-022-33963-w). The authors shortly mention the importance of H2O2 briefly in their text (Line 87).

Response: Yes, when H2O2 is used as a co-substrate, the LPMO reaction is faster. In this study, we have focused on the characteristics of PcCDH and its ability to act as an electron donor of LPMO, therefore, we haven’t investigated the effect of different co-substrates on LPMO’s reaction.

Figure 3: What is the difference between the samples in lane 2 and 3 or lane 4 and 5, respectively? Is the protein expressed under the control of different promoters applied twice?

Response: We have used both the pdc promoter and the cDNA1 promoters to drive PcCDH expression in T. reesei, and found that pdc promoter is more effective. However, to avoid misunderstanding, we have deleted the contents related with the cDNA1 promoter.

Please also correct “KDa” to “kDa”.

Response:  We have modified it.

The authors should show an SDS-PAGE result to confirm the purity of their protein, not only a western blot. Just from the Western blot it is not possible to prove if the protein is pure or not.

Response:  Previously we added western blot analysis of PcCDH expressed under two different promoters. However, we have now repeated the SDS-PAGE analysis for PcCDH expressed under pdc promoter and removed the western blot analysis.

Line 131.: The authors found 2 distinct bands of their target protein by Western blot analysis. The argue that this is due to different glycoforms. However, in my opinion the bands are too sharp to account for two distinct glycoforms. Form other studies it is well known that CDH can be proteolytically cleaved into the DH and the CYT domain. The isolated DH domain is often found as when purifying CDHs from native fungi as well as expression hosts. In a previous paper PcCDH was also heterologously expressed in T. reesei and the authors reported to obtain the DH domain as well as the full-length CDH. (https://doi.org/10.1186/s12934-020-01492-0). The size of the lower band corresponds very well to the size of the DH domain and the results presented by the authors here are very similar to the observations made in the paper mentioned before. The authors need definitely to consider this and check the possibility of the presence of the DH domain.

Response: We have identified both the bands through mass spectrum analysis, and results show that both the bands are full length PcCDH. Therefore, we assume that the difference in their molecular weight is caused by the difference in glycosylation. We have added the results of mass spectrum analysis in the revised manuscript.

Figure 4: Which assay was used for the pH-optimum? DCIP as electron accepter? Cellobiose as electron donor? Which concentrations? Please provide this information in the Figure’s caption.

Response: The condition for pH-optimum assay is described in section 4.5. We have provided this information in the figure’s caption in the reversion.

Figure 5: Why is the absorbance of the protein so high at 280 nm compared to the absorbance of the heme at 420 nm? Native PcCDH has a ratio of A420/A280 of > 0.6 (https://doi.org/10.1016/S0167-4838(97)00180-5) and the same is true for heterologously produced PcCDH (https://doi.org/10.1186/s12934-020-01492-0). The spectrum reported here shows an A420/A280 of < 0.3 which would mean that the protein purity is very low or that the heme loading is insufficient. The authors definitely need to comment on that.

Response: Yes, it’s a concern. Because we used a high glucose concentration medium for recombinant PcCDH production, thus, the expression of native hydrolases of T. reesei is effectively repressed as shown in Fig. 3. Therefore, contamination of other protein in the sample is almost impossible. Possibly due to insufficient heme loading. We are re-cultivated the recombinant T. reesei strain. However, due to the shortage of time for the revision submission, we haven’t made any change regarding this issue in the revised manuscript.

Figure 6: What do the authors measure at 560 nm? Please explain in the figure caption. Furthermore, the authors need to check the purity of their preparations (especially xylan and gum Arabic) as the Km for cellobiose and other oligosaccharides is very low, already small contents of these sugars would lead to false positive results.

Response: In this Figure, we used the Amplex red-horseradish method to measure the production of H2O2, and we have described it in the figure caption of revised manuscript. Xylan and gum Aribic are commercially purchased from Sigma Aldrich and Sangon respectively.

Figure 7: How much cellobiose was added in the second sample? I cannot find 7a and 7b here but only one plot with two samples. Please check and correct description.

Response: We used 1 mM cellobiose. We have marked it in red color in section 2.4 and corrected the description of the figure caption.

Figure 8: Also, here the Figure caption needs improvement. Which assay was used? Turbidimetry?

Response: Yes, we assayed the consumption of PASC using the turbidimetry method. We have rewritten the figure caption.

The time scale between the experiments presented in Figure 8 (72 h) and Figure 9 (15 min) are completely different. The authors definitely need to explain better why they see the boosting effect of LPMO only after more than 24 h while inactivation of LPMO seems to occur already much faster (min – h scale)

Response: In Fig. 9, we compared the deactivation rate of LPMO with the application of different external electron donors. We found that ascorbic acid rapidly deactivates LPMO in the absence of PASC. Therefore, the duration in Fig. 9 is short. However, when PcCDH is used as an electron donor alone, LPMO remained comparatively active for a longer period. We have modified Fig. 9 in the revised version.

Minors:

Line 27f.: This sentence is not clear to me. Is there a mistake in the sentence?

Response: Yes, there was a mistake and we have removed it.

Line 44f. I think it would be too generic to state that the specific function of oxidative enzymes of fungi is “elusive” There are a lot of studies targeting this topic and reasonable models for the function of many oxidoreductases in lignocellulose deconstruction.

Response: We have removed this sentence.

Line 53: Do the authors mean longer cellooligosaccharides instead of “longer cellodextrin”?

Response: Yes, we have modified it as cello-oligosaccharides.

Line 58: Grammar and style need to be improved

Response: We have improved the grammar and style.

Line 62: Quinones can also be reduced at the isolated DH domain (e.g. 1,4-benzoquinone, 2,6- dimethyl-1,4-BQ, etc.). The cytochrome domain is not involved in their reduction. The same is true for DCIP while cytochrome c can only be reduced via the cytochrome domain.

Response: We have modified it accordingly.

Line 64: Replace “Cytochrome C” by “cytochrome c”

Response: We have replaced it.

Line 67: Regarding LPMO classification the authors should also include recent literature and the addition of another family of LPMOs (AA17).

Response: We have added the latest information.

Line 68: Grammar needs to be corrected (…belongs…) and language and style need to be improved.

Response: We have modified this sentence

Line 69. The authors write: “These LPMOs in synergism with cellulases can oxidatively cleave crystalline cellulose through hydrolysis”. This is misleading as LPMO is not performing hydrolysis but oxidative cleavage of the carbohydrate backbone.

Response: We have modified this sentence

Line 80f.: The authors use the “terms inter-domain electron transfer” (IET), “direct electron transfer (DET)” and “electron transfer” not correctly. IET is defined as the electron transfer form the DH to the CYT domain and not to an “external electron acceptor”. DET is defined as the electron transfer from the enzyme to an electrode without mediator.

Response: We have modified this part.

Line 96: grammar (loses instead of losses).

Response: We have modified this sentence accordingly.

Line 127f. The authors just measured the total protein concentration to identify the best producer. What is the proportion of CDH relative to the remaining proteins in the culture supernatant? Why are the authors not using a CDH-specific activity assay (DCIP, etc.)?

Response: According to the SDS-PAGE analysis, the recombinant PcCDH is dominant in the culture supernatant, so we measured that total protein concentration to identify the best produce. We used DCIP as the electron acceptor in some assays such as the kinetic parameters assay and the pH optimum assay.

Line 152: change “kinetic parameters” to “kinetic constants”

Response: We have changed it

Line 154f.: The authors write: “PcCDH has a lower affinity for the substrate (lower Km), more catalytic for the bound substrate (higher kcat). Similarly, PcCDH has a higher catalytic constant (kcat/Km) for fibrous oligosaccharides.” – This is absolutely not clear for me and I think needs proper revision. Furthermore, what do the authors mean by “fibrous oligosaccharides”? Cellooligosaccharides?

Response: We have rewritten this part in the reversion. We have corrected the mistake “fibrous oligosaccharides” to “cello-oligosaccharides”.

Table 1: correct “Kcat” to “kcat”.

Response: We have modified it

Table1: The authors use to many significant digits (especially for kcat values) please check and apply correct rounding of measurement results.

Response: We have modified it (used two decimals)

Line 359: What is the ascorbate oxidase used for in this study?

Response: Ascorbate oxidase is used to oxidize ascorbic acid for LPMO activity assay.

Line 385, Line 388: Please change uL to “µL” and “ug/mL” to “µg/mL”

Response: We have changed uL to “µL” and “ug/mL” to “µg/mL”

Line 397: correct “ÄKTA”

Response: We have corrected it

Line 480: What do the authors mean by cellulose disaccharides or cellulose trischaccharides?

Response: We apologize for the mistake. We used cellobiose and cellotriose.

There needs to be space between number and unit. Please correct. E.g. Line 318, Line 469, Line 474, Line 475, Line 476, Line 448, etc.

Response: We have added proper spacing between the lines

Round 2

Reviewer 1 Report

Comments and Suggestions for Authors

The authors have addressed the reviewers' comments and improved the manuscript. Therefore, the manuscript is acceptable.

Author Response

We are really thankful to you for your time and effort in reviewing our manuscript and considering it for publication in IJMS.

Reviewer 2 Report

Comments and Suggestions for Authors

The authors have responded to all the points I raised. Although I am still not entirely convinced of the novelty of the expression and characterisation of PcCDH, the authors have correctly pointed out that the novelty lies in the interaction with LPMO. Indeed, the function as an electron donor for LPMO of this particular enzyme represents new and larger part of the study presented. I further appreciate that the authors now present a proof of purity by SDS-PAGE as well as the verification of the obtained protein by MS analysis. The manuscript has been greatly improved, but I have a few more comments that need to be discussed or corrected:

Line 55: Please correct “dehydrogenation domain (DH)” to “dehydrogenase domain (DH)”

Line 64: the “c” in cytochrome c should be written in italics

Figure 1: I agree with the authors decision to remove the structural model. In the reaction scheme, however, the authors still focus on the oxygen-dependent mechanism of the LPMO reaction. Is there a reason, why the authors use the term “PMO” in the scheme? In the remaining manuscript they always use LPMO. This should be checked or argued why they use different abbreviations.

Figure 4: Spectral analysis. As I have outlined in my first review of the manuscript the RZ value (A420/A280) is a characteristic value for the purity of CDH. As the authors have argued an insufficient heme loading could be an explanation for the low value found in this study (although the specific activity seems to be very high). In my opinion the authors would definitely need to add some discussion regarding this topic into the manuscript.

Figure 5: Also, the findings regarding the substrate specificity need some more discussion (especially why the activity on gum Arabic is that high. Is it because this substrate contains small amounts of cellobiose, glucose or cellooligosaccharides?)

Line 190: I guess the authors mean heme b cofactor and not “haemoglobin b cofactor”

Line 201: Change H2O2 to H2O2

Line 202: Change “100mM” to “100 mM”

Line 356: Please correct “Camella” to “Cannella”

Line 511ff.: Please do not capitalize species names. Only genus names should be capitalized.

Comments on the Quality of English Language

Minor editing is required

Author Response

The authors have responded to all the points I raised. Although I am still not entirely convinced of the novelty of the expression and characterisation of PcCDH, the authors have correctly pointed out that the novelty lies in the interaction with LPMO. Indeed, the function as an electron donor for LPMO of this particular enzyme represents new and larger part of the study presented.

I further appreciate that the authors now present a proof of purity by SDS-PAGE as well as the verification of the obtained protein by MS analysis. The manuscript has been greatly improved, but I have a few more comments that need to be discussed or corrected:

Response:

We express our sincere gratitude for your insightful evaluation of our work, which ultimately guided us in making comprehensive improvements.

Line 55: Please correct “dehydrogenation domain (DH)” to “dehydrogenase domain (DH)”

Response:

We have modified it.

Line 64: the “c” in cytochrome c should be written in italics

Response:

We have italicized it.

Figure 1: I agree with the authors decision to remove the structural model. In the reaction scheme, however, the authors still focus on the oxygen-dependent mechanism of the LPMO reaction. Is there a reason, why the authors use the term “PMO” in the scheme? In the remaining manuscript they always use LPMO. This should be checked or argued why they use different abbreviations.

Response:

We have modified the reaction scheme and used the term LPMO instead of PMO.

Figure 4: Spectral analysis. As I have outlined in my first review of the manuscript the RZ value (A420/A280) is a characteristic value for the purity of CDH. As the authors have argued an insufficient heme loading could be an explanation for the low value found in this study (although the specific activity seems to be very high). In my opinion the authors would definitely need to add some discussion regarding this topic into the manuscript.

Response:

As the kinetic parameters and specific activity of our recombinant PcCDH align with those documented in the literature, we presume that both bands observed in the SDS-PAGE represent purified PcCDH, whether glycosylated or unglycosylated, and possess full activity. The lower A280/A420 value in our recombinant PcCDH could potentially be attributed to insufficient binding of the heme b cofactor. The rapid synthesis of recombinant PcCDH is regulated by the robust pdc promoter, whereas the production of heme b may be comparatively limited. We have incorporated discussions on this aspect in the revised manuscript.

Figure 5: Also, the findings regarding the substrate specificity need some more discussion (especially why the activity on gum Arabic is that high. Is it because this substrate contains small amounts of cellobiose, glucose or cellooligosaccharides?)

Response:

We have detected the reducing sugars in the gum Arabic, and to avoid misunderstanding, we have removed the results of gum Arabic in the revised manuscript.

Line 190: I guess the authors mean heme b cofactor and not “haemoglobin b cofactor”

Response:

We apologize for the mistake. We have corrected it.

Line 201: Change H2O2 to H2O2

Response:

We have modified it.

Line 202: Change “100mM” to “100 mM”

Response:

We have modified it.

Line 356: Please correct “Camella” to “Cannella”

Response:

We apologize for the mistake. We have removed this typing error.

Line 511ff.: Please do not capitalize species names. Only genus names should be capitalized.

Response:

We extremely regret and apologize for such a grave mistake. We have edited all the species names. 

Comments on the Quality of English Language: Minor editing is required

Response:

We have edited the language.